# Enhancing Gamma-Ray Burst Detection: Evaluation of Neural Network Background Estimator and Explainable AI Insights

Riccardo Crupi [1,2,*] , Giuseppe Dilillo [3] , Giovanni Della Casa [3] , Fabrizio Fiore [2] and Andrea Vacchi [1,4]

1. Dipartimento di Scienze Matematiche, Informatiche e Fisiche, Università di Udine, Via delle Scienze 206, 33100 Udine, Italy
2. Osservatorio Astronomico di Trieste, INAF, Via Tiepolo 11, 34143 Trieste, Italy
3. IAPS, INAF, Via del Fosso del Cavaliere 100, 00133 Roma, Italy
4. Sezione di Trieste, INFN, Via Padriciano 99, 34149 Trieste, Italy
* Correspondence: crupi.riccardo@spes.uniud.it

**Abstract:** The detection of Gamma-Ray Bursts (GRBs) using spaceborne X/gamma-ray photon detectors depends on a reliable background count rate estimate. This study focuses on evaluating a data-driven background estimator based on a neural network designed to adapt to various X/gamma-ray space telescopes. Three trials were conducted to assess the effectiveness and limitations of the proposed estimator. Firstly, quantile regression was employed to obtain an estimation with a confidence range prediction. Secondly, we assessed the performance of the neural network, emphasizing that a dataset of four months is sufficient for training. We tested its adaptability across various temporal contexts, identified its limitations and recommended re-training for each specific period. Thirdly, utilizing Explainable Artificial Intelligence (XAI) techniques, we delved into the neural network output, determining distinctions between a network trained during solar maxima and one trained during solar minima. This entails conducting a thorough analysis of the neural network behavior under varying solar conditions.

**Keywords:** GRB; neural network; XAI

## 1. Introduction

The rapid identification of various transient types, such as Gamma-Ray Bursts (GRBs), is crucial, especially when considering multiple observations related to multi-wavelength event analysis. For instance, it plays a key role in alerting ground telescopes for pointing direction and subsequent afterglow observations, facilitating follow-up studies. GRBs are distinguished by increases in detector count rates, indicating activity that appears unusual and cannot be explained by background or known sources. Real-time modeling of the physical background is inherently difficult, so algorithms dedicated to "online" GRB detection can extrapolate the background from recent observations, such as from the onboard detector systems Fermi GBM [1], Compton-BATSE [2] and BeppoSAX-GRBM [3].

In "offline" analysis, researchers search archival data for GRB events that may have gone unnoticed. Examples of this methodology can be found in studies such as Kommers et al. (1999) [4], which uses the BATSE catalog, or in Kocevksi et al. (2018) [5] and Hui et al. (2017) [6], where the search focuses on faint, short GRBs that occur concurrently with known gravitational wave events.

Biltzinger et al. (2020) [7] introduce a novel approach that provides comprehensive estimation by incorporating detailed models catching various background factors anticipated for the Gamma-Ray Burst Monitor (GBM). These factors encompass detector response, cosmic-ray background, solar activity, geomagnetic environment, Earth albedo and the visibility of X and gamma-ray point sources. The resulting background estimation closely replicates Fermi GBM observations, particularly the long temporal variations (e.g., 10 min), suggesting potential applicability in detecting challenging GRBs, such as long weak events

with slow rise times. However, because this technique is specifically designed for Fermi GBM observations, its application to other experiments is limited.

The use of deep learning approaches provides a promising solution for improving the accuracy and efficiency of detection of GRBs or other X/gamma-ray astrophysical transients.

Sadeh (2019) [8] introduces an alternative method for background prediction based on a recurrent neural network (RNN). This RNN, which was trained on existing burst observation catalogs to identify GRB events, raises concerns about the potential transfer of detection biases from traditional GRB detection strategies. This disadvantage may result in overlooking events that previous search methodologies had missed.

In a related context, Crupi et al. [9,10] describes a feedforward neural network (FFNN) for estimating X/gamma-ray backgrounds for space telescope detectors. This method, which is accompanied by an efficient changepoint detection technique [11,12], takes advantage of all available satellite information (variables), as employed in [7]. Importantly, because the learning is data-driven, it can be applied to any X/gamma-ray photon detector on a space satellite and does not require event exclusion during training. The following section describes two experiments designed to test the robustness of this method, identify its limitations and improve the interpretation of its output, taking into account the neural network inherent "black box" nature.

The neural network was tested using Fermi GBM data, but the problem formulation is intended to be adaptable to other satellites, including upcoming ventures such as the technological and scientific pathfinders HERMES-TP (funded by ASI) and HERMES-SP [13–17] (funded by the European Commission, collectively known as HERMES Pathfinder). The primary goal of HERMES Pathfinder is to demonstrate the feasibility of detecting and localizing GRBs using miniaturized instrumentation hosted by nanosatellites. The first six HERMES Pathfinder spacecraft are set to launch into low-Earth, near-equatorial orbits in 2024/2025.

## 2. Methods

### 2.1. Data

The Fermi GBM [1] detectors considered are the twelve NaI photomultipliers: n0, n1, n2, n3, n4, n5, n6, n7, n8, n9, na and nb. In the context of this study, which focuses on the detection framework [9] pertaining to long transients, the count rate light curves are binned in time with 4.096 s binlength. Moreover, the energy range is binned in three categories: 28–50 keV ($r0$), 50–300 keV ($r1$) and 300–500 keV ($r2$). In total, there are 36 variables that are called *col_range* (range columns): 12 count rates (in 4.096 s) times 3 energy range; see Table 1.

With the Fermi Data Tools package [18], it is possible to retrieve information of the satellite in a particular timestamp. There are 24 of these variables (orbital information of Fermi, Earth and Sun location, etc.), and they are referred to as *col_sat_pos* (satellite position columns); see Table 2.

Information relative to the detector is also available. In total, there are 36 variables (pointing of the detectors and FOV Earth occlusion), and they are called *col_det_pos* (detector position columns); see Table 3.

**Table 1.** The table consists of 36 detector features aligned with the target data table, with the detector label denoted as $i \in \{0, 1, 2, 3, 4, 5, 6, 7, 8, 9, a, b\}$. The target data represent count rates binned at 4.096 s, sourced from the daily CSPEC files and processed using the Fermi GBM Data Tools library.

| Target Label | Description |
|---|---|
| n$i$_r0 | $i$-labeled detector count rates in range $r0$ |
| n$i$_r1 | $i$-labeled detector count rates in range $r1$ |
| n$i$_r2 | $i$-labeled detector count rates in range $r2$ |

**Table 2.** A table containing 24 orbital features, derived from the POSHIST files and processed using the Fermi GBM Data Tools library, serves as the input for the neural network (NN). These features pertain to satellite position, velocity, Sun and Earth visibility, passage through the South Atlantic Anomaly (SSA) and the local McIlwain L value (*l*), which is determined based on Fermi orbital position and corresponds to a geomagnetism parameter that describes the magnetic field intensity along a specific magnetic field line in the Earth's magnetosphere. Reproduced with permission from [9] ©Crupi et al. (2023), CC BY 4.0.

| Feature Label | Description |
|---|---|
| *pos_x*, *pos_y*, *pos_z* | position of Fermi in Earth inertial coordinates |
| *a, b, c, d* | Fermi attitude quaternions |
| *lat* | Fermi geographical latitude |
| *lon* | Fermi geographical longitude |
| *alt* | Fermi orbital altitude |
| *vx, vy, vz* | velocity of Fermi in Earth inertial coordinates |
| *w*1, *w*2, *w*3 | Fermi angular velocity |
| *sun_vis* | Sun's visibility boolean flag |
| *sun_ra* | Sun's right ascension |
| *sun_dec* | Sun's declination |
| *earth_r* | Earth's apparent radius |
| *earth_ra* | Earth center right ascension |
| *earth_dec* | Earth center declination |
| *saa* | SAA transit boolean flag |
| *l* | approximate McIlwain L value |

**Table 3.** A table comprising 36 detector features is utilized to construct the neural network input table, with the detector label denoted as $i \in \{0, 1, 2, 3, 4, 5, 6, 7, 8, 9, a, b\}$. These features are extracted from the POSHIST files and processed using the Fermi GBM Data Tools library. The detector features encompass pointing coordinates in equatorial coordinates and indicate whether a detector Field of View (FOV) is occulted by the Earth. Reproduced with permission from [9] ©Crupi et al. (2023), CC BY 4.0.

| Feature Label | Description |
|---|---|
| n*i_ra* | *i*-labeled detector pointing right ascension |
| n*i_dec* | *i*-labeled detector pointing declination |
| n*i_vis* | *i*-labeled detector Earth occultation boolean flag |

*2.2. Problem Setting*

The problem is formulated as a typical supervised machine learning estimation, where inputs *X* are the variables in *col_sat_pos* and *col_det_pos* and the outputs *Y* are the variables in *col_range*. The model, denoted as *F*, is trained to provide an estimation such that $Y \approx F(X)$. In this multi-output regression, the model *F* approximates the background photon count rates based on the variables of the satellite and its detectors. The model employed is a feed forward neural network with 3 hidden dense layers and 3 dropouts, 1 per each hidden layer; see Figure 1.

During the pre-processing phase, the input training dataset is standardized, and data collected while Fermi passes through the high radiation environment of the South Atlantic Anomaly (SAA) are removed. The SAA, characterized by reduced geomagnetic intensity, facilitates the closer penetration of charged particles from space to the Earth's surface. These particles interact with the detectors, increasing the background count rate and posing challenges to accurate estimation. Consequently, data collected within the SAA are excluded to ensure the reliability of estimation.

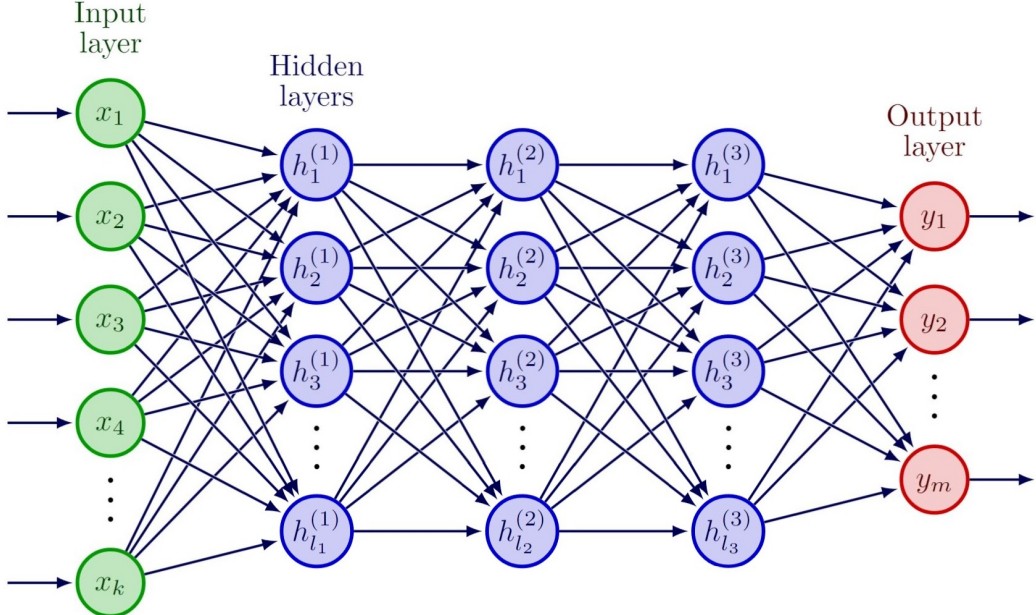

**Figure 1.** The architecture of a feed forward neural network. The input has a dimension of 60. The first two hidden layers have 2048 neurons, the third 1024. The output layer has a dimension of 36.

The splitting procedure divides the dataset into train and test sets; part of the training set is further kept as the validation set. The loss function is the Mean Absolute Error

$$\text{MAE}(y, \hat{y}) = \frac{1}{n} \sum_{i=1}^{n} |y_i - \hat{y}_i| \tag{1}$$

where $n$ is the total number of samples in the training set, $i$ refers to the specific sample, $y_i$ the target value (the observed count rate) and $\hat{y}_i$ is the estimated value (the estimated count rate).

More generally, $|y_i - \hat{y}_i|$ in Equation (1) can be substituted to

$$L_\tau(y, \hat{y}) = \begin{cases} \tau(y - \hat{y}) & \text{if } y \geq \hat{y}, \\ (\tau - 1)(y - \hat{y}) & \text{if } y < \hat{y}, \end{cases} \tag{2}$$

to form the quantile loss, and if $\tau = 0.5$, then it is equivalent to MAE.

In the settings of multi-output regression, the overall loss is

$$L = \min_{i}(\text{MAE}(f_j(X), Y_j)) \tag{3}$$

where $j$ spans *col_range*.

For a more robust evaluation of the error, the Median Absolute Error is employed:

$$\text{MeAE}(y, \hat{y}) = \text{median}(| y_i - \hat{y}_i |) \tag{4}$$

where the terms are the same as in Equation (1).

## 3. Results

### 3.1. Quantile Regression

By employing the quantile loss function, three neural networks were trained, each targeting a different quantile value: 0.1, 0.5 and 0.9. When $\tau = 0.1$, the network estimates the 10th percentile of the distribution $\mathbb{P}(Y \mid X = x)$. For $\tau = 0.5$, the network provides the median estimation, which is equivalent to minimizing the Mean Absolute Error (MAE). Finally, when $\tau = 0.9$, the network estimates the 90th percentile.

Quantile regression offers a range of confidence for count rate predictions, as illustrated in Figure 2. For the trigger algorithm, to enhance robustness in detection, the prediction with $\tau = 0.9$ could serve as an estimate of the true count rates.

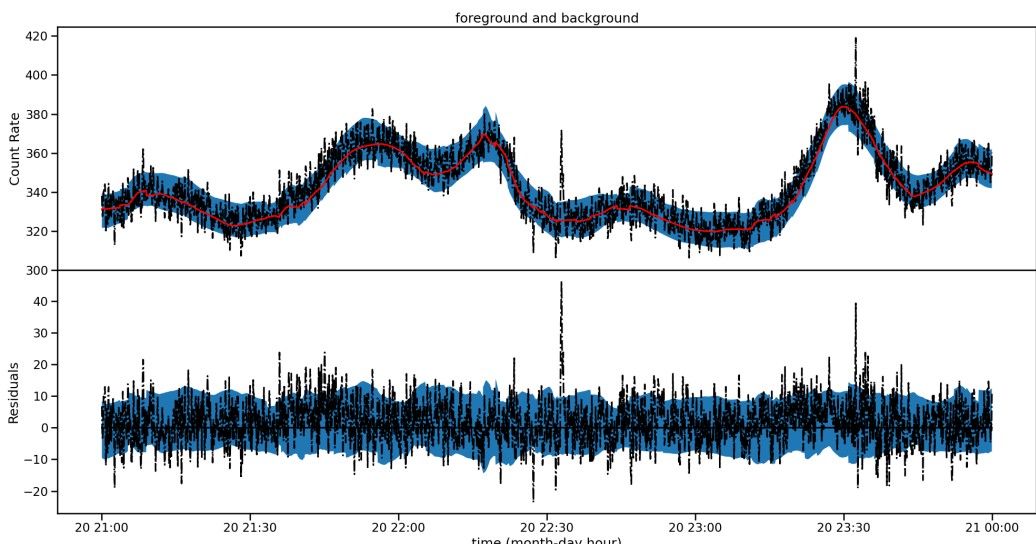

**Figure 2.** Quantile regression for estimating the count rates of detector *n*7 in energy range *r*1 over a three-hour period on 20 April 2019. The red line represents the $\tau = 0.5$ (median) estimate, while the blue shaded region denotes the range between the upper value at $\tau = 0.9$ and the lower value at $\tau = 0.1$. The spike observed at 23:30 corresponds to GRB190420981, whereas the one at about 22:30 is not present in the Fermi GBM catalog but is detected by [9] and further discussed.

### *3.2. Data Quantity and Out-of-Time Test Set Evaluation*

Once the model is trained, questions arise regarding (1) the amount of data required for the neural network to converge and achieve good performance. Additionally, it is interesting to understand (2) the network behavior when applied to data outside the training period.

To address the first question, thirteen experiments were conducted within the data period spanning 2019 to 2020, characterized by solar minima to minimize the impact of solar X/gamma-rays and flares. Table 4 displays results for training periods of 4, 7, 12 and 24 months, using 75% of the dataset for training (30% of which was further reserved as a validation set) and 25% for testing. To facilitate convergence, a dropout of 0.002 was chosen, and the learning rate for the 24-month dataset was reduced by a factor of 10 compared to [9]. The performance metrics, both MAE and MeAE, show no significant improvement with an increase in the dataset size. The performance remains consistently good and comparable beyond a 4-month training period. For reference, one month of data comprises approximately 500,000 datapoints.

To address the second question, two group of experiments were conducted on data covering the period of 2019 and 2020. Experiments # 6 to # 9 reserve a "future" period as a test set with respect to the train set, while # 10 to #13 display the performance of an NN trained on data "in the middle" with respect to the train set. In particular, experiment # 6 reserves as the test set July 2020 to January 2021, and # 10 displays the performance of an NN trained on data excluding January 2020 to July 2020 (6 months or 25% of the dataset). Training results align with previous experiments, as expected since the data are an extension of the previous dataset. However, applying the model to an out-of-time period results in slightly poorer performance compared to the in-time sets, with an MAE increase of approximately 2.5 counts/s and a MeAE increase of 1.0 counts/s. Considering that the average background rate is around 350 counts/s (as shown in Figure 2), this corresponds to a relative percentage error increase of about 0.7% and 0.3%, respectively. This indicates

that while the NN did not perfectly generalize to a period outside the training set, it still yielded comparable results. Experiments # 7, # 8, # 9 are run over an increasing size of the train set but tested on the same 2 months of data, Nov20–Jan21. We can see that the MAE is even higher than that of # 6, and a closer examination of the MAE reveals a more pronounced impact on the *r*0 range, which is most susceptible to solar activity. This suggests that including a temporal feature to account for the varying effects of solar activity on the detectors would be beneficial over longer time periods. The number of solar flares, thus a proxy for solar activity, can be seen in Figure 3. Finally, experiments # 11, # 12, # 13 are the equivalent of the previous experiments but tested on a test set "in the middle", Jan20–Mar20, which shows slightly better performance on the test set and is not influenced by the train set size.

It is important to note that the actual purpose of the NN is not to predict the future; thus, changes in architecture, such as RNN, may be beneficial for more accurate predictions. The NN serves as a tool to interpolate expected count rates for chosen periods, demonstrating robustness against unexpected increases in count rates, such as those caused by solar flares or GRBs.

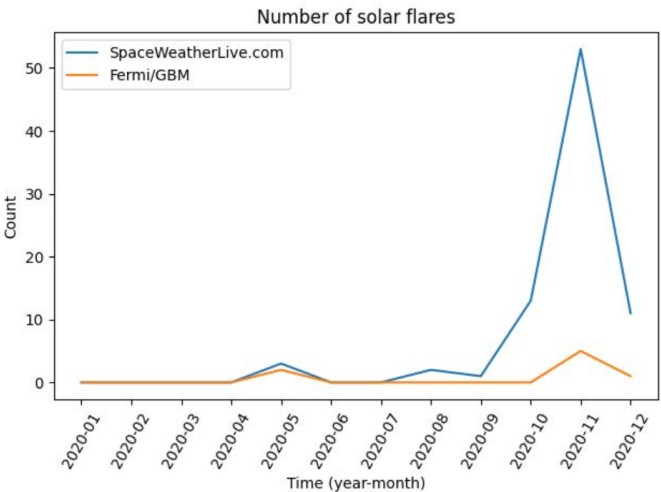

**Figure 3.** Solar flares counted by GOES-X satellite (source: spaceweatherlive.com, accessed on 10 January 2024) and by the Fermi GBM (source: Fermi GBM trigger catalog).

**Table 4.** For each training period, spanning from the initial month to the end month (excluding), a neural network was trained, and the resulting MAE and MeAE metrics are reported, averaged per detector and energy range. In some experiments, the total duration of the dataset was delineated as the sum of the training months, validation months and test months. The first five rows provide a comparison between training and test results, illustrating that the neural network avoids overfitting when the test set is randomly selected within the same period as the training set (in-time). However, experiments from # 6 to # 13 reveal that the neural network does not perfectly generalize background estimation when applied to a period not included in the training set (out-of-time). Nonetheless, the NN still yields satisfactory results in these out-of-time scenarios. Here, "#" stands for the specific experiment number.

| Exp # | Total Months | Train Period | Test Period | MAE Train (Counts/s) | MAE Test (Counts/s) | MeAE Train (Counts/s) | MeAE Test (Counts/s) |
|---|---|---|---|---|---|---|---|
| 1 | 4 | Mar19–Jul19 | in-time | 4.383 | 4.421 | 3.361 | 3.370 |
| 2 | 7 | Jan19–Jul19 | in-time | 4.280 | 4.306 | 3.296 | 3.310 |
| 3 | 12 | Jan19–Jan20 | in-time | 4.446 | 4.442 | 3.353 | 3.361 |
| 4 | 12 | Jan20–Jan21 | in-time | 4.202 | 4.222 | 3.314 | 3.327 |
| 5 | 24 | Jan19–Jan21 | in-time | 4.445 | 4.449 | 3.421 | 3.423 |

**Table 4.** *Cont.*

| Exp # | Total Months | Train Period | | Test Period | MAE Train (Counts/s) | MAE Test (Counts/s) | MeAE Train (Counts/s) | MeAE Test (Counts/s) |
|---|---|---|---|---|---|---|---|---|
| 6 | (13 + 5) + 6 | Jan19–Jul20 | | Jul20–Jan21 | 4.296 | 6.684 | 3.335 | 4.532 |
| 7 | (16 + 2) + 2 | Jan19–Jul20 | | Nov20–Jan21 | 4.888 | 7.518 | 3.744 | 4.543 |
| 8 | (18 + 2) + 2 | Jan19–Aug20 | | Nov20–Jan21 | 4.464 | 7.844 | 3.403 | 4.790 |
| 9 | (20 + 2) + 2 | Jan19–Nov20 | | Nov20–Jan21 | 4.256 | 7.790 | 3.321 | 4.681 |
| 10 | (13 + 5) + 6 | Jan19–Jan20 Jul20–Jan21 | and | Jan20–Jul20 | 4.390 | 6.980 | 3.365 | 4.626 |
| 11 | (16 + 2) + 2 | Jan19–Jan20 May20–Jan21 | and | Jan20–Mar20 | 4.536 | 7.105 | 3.418 | 4.724 |
| 12 | (18 + 2) + 2 | Jan19–Jan20 May20–Jan21 | and | Jan20–Mar20 | 4.787 | 6.667 | 3.631 | 4.778 |
| 13 | (20 + 2) + 2 | Jan19–Jan20 Mar20–Jan21 | and | Jan20–Mar20 | 4.910 | 6.609 | 3.707 | 4.915 |

*3.3. Xai Application*

XAI (Explainable Artificial Intelligence) techniques are methodologies employed in artificial intelligence to enhance the transparency and interpretability of machine learning models. They offer explanations for the decisions made by AI systems, particularly in complex tasks where the rationale behind the output might not be obvious. By facilitating understanding of AI model reasoning, XAI techniques foster trust, accountability and the identification of biases or errors in AI systems.

Morris sensitivity analysis [19] is a one-step-at-a-time global sensitivity analysis method employed to quickly screen important inputs in a model. However, it does not account for nonlinear interactions, making it suitable for initial input screening but limited in capturing complex relationships. On the other hand, Kernel SHAP (Shapley Additive Explanations) [20] is a powerful XAI technique that quantifies each feature contribution to a model output. Based on cooperative game theory concepts, Kernel SHAP provides a unified framework for feature importance analysis by calculating the average contribution of each feature across various coalitions. This technique offers a comprehensive understanding of individual feature contributions, enhancing the interpretability of complex models and aiding in model debugging.

XAI techniques such as Morris sensitivity analysis and Kernel SHAP were used to better understand and improve the NN.

3.3.1. GlobalExplanation

To identify the most influential features and their impact on predictions, a Morris sensitivity analysis was performed using the InterpretML library [21]. The choice for this method was due to its computational efficiency, as it perturbs one feature at a time. However, it does not consider non-linearities when estimating output, which it is addressed later in local explanations for individual datapoints. Since the NN produces 36 outputs, feature importance is sorted based on the average importance. Figures 4 and 5 illustrate this for each detector/range combination. We applied this analysis to NNs trained on data from the 2019 and 2014 periods, corresponding to solar minima and solar maxima, respectively.

In both figures, Fermi geographical longitude and the McIlwain L-parameter emerge as the most significant features. However, there is a notable difference in Figure 5, where the Earth's occultation of detectors exhibits high significance. The discrepancy between the results in Figures 4 and 5 can be attributed to the variance in the underlying data distribution and characteristics between the two periods. For instance, during periods of high solar activity, such as 2014, the obscuration of a detector's Field of View (FoV) by the Earth could lead to fewer count rates: the Earth blocks the line of sight between the detector and the source of radiation. Conversely, in 2019, with significantly less solar noise (e.g., fewer solar flares), the importance of these features becomes negligible. This suggests that

training a neural network for each period could yield benefits, as the model could leverage different features to reveal distinct patterns in the data.

Further exploration of the detector flags indicating the Earth's FoV occultation is undertaken through an example of local analysis in the following section.

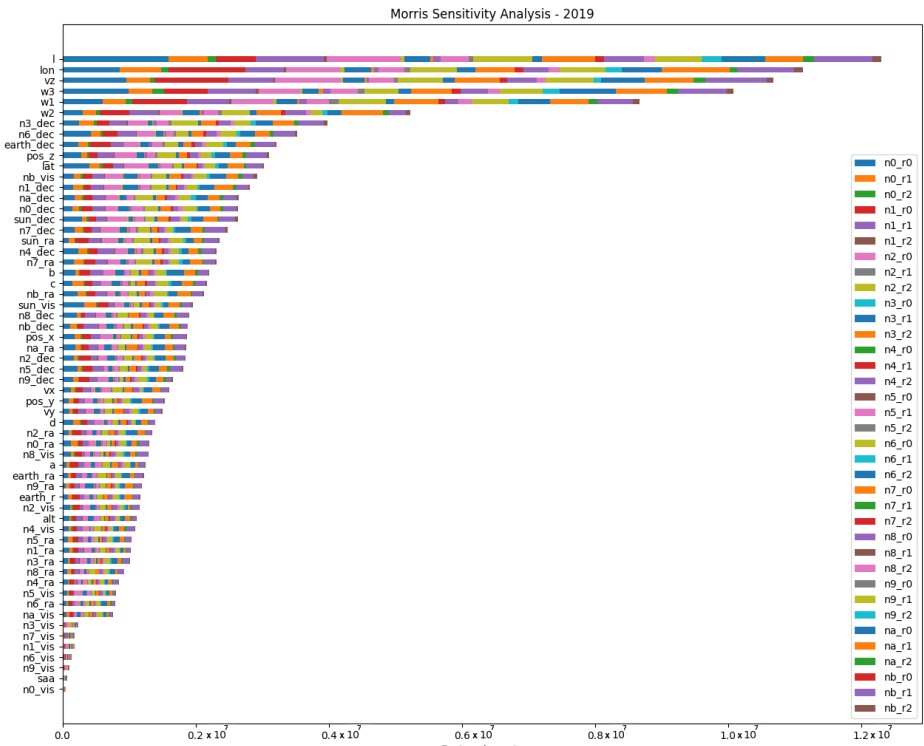

**Figure 4.** Feature importance as determined by the Morris method for the 2019 dataset. Features are ranked by average importance, with the McIlwain L-parameter and Fermi longitude coordinate at the top. Conversely, certain detector flags indicating the Earth's FoV occultation emerge as almost irrelevant factors in prediction. For a complete list of features and their descriptions, refer to Tables 2 and 3.

### 3.3.2. Local Explanation

Local explanations focus on individual datapoints, and for this purpose, Kernel SHAP was utilized. It allows for the generation of explanations that cover the entire NN, providing a feature importance list for each of the 36 outputs and aiding in debugging unusual predictions. For instance, Figure 6 showcases an unexpected peak for detector n8 in energy range r1. Kernel SHAP in Figure 7 highlights the importance of the Fermi geographical longitude, which is expressed in degrees: it gradually increases to $360°$ and then returns to $0°$ as expected. However, due to a dataset issue (an interpolation error in longitude estimation), the value jumps to around $100°$, invalidating the result. The solution involves rectifying the dataset, as the NN strives to provide the best estimation possible.

Nevertheless, this specific behavior has minimal impact on the application of the trigger algorithm employed in [9]. The error is confined to a single datapoint, and the trigger algorithm is sensitive to excessive count rates and not to lower count rates.

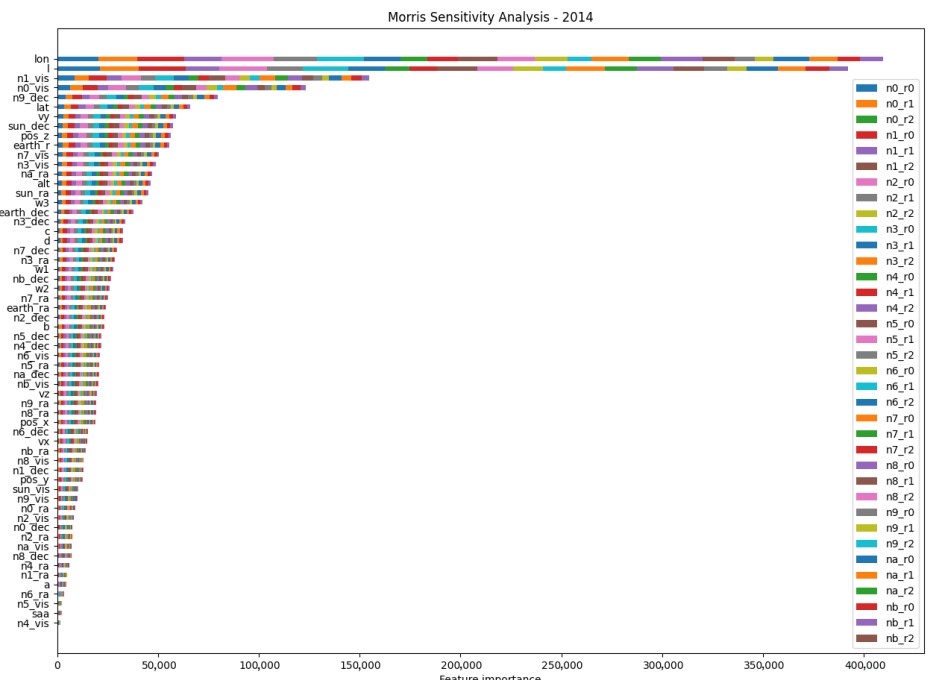

**Figure 5.** Feature importance as determined by the Morris method for the 2014 dataset. Features are sorted by their average importance, with the McIlwain L-parameter and Fermi longitude coordinate as the most influential. In contrast to Figure 4, detector flags representing the Earth's FoV occultation play a significant role in prediction. For a complete list of features and their descriptions, refer to Tables 2 and 3.

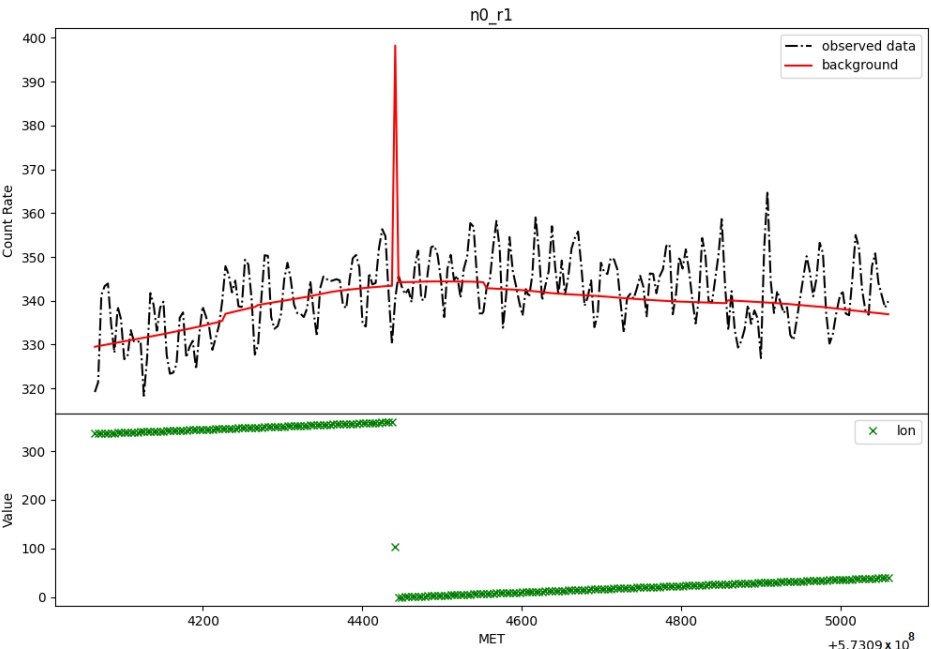

**Figure 6.** Light curve of observed and estimated count rates for detector n0 in energy range *r*1. An unexpected peak appears in the estimated count rates due to an error in the dataset for the longitude estimation.

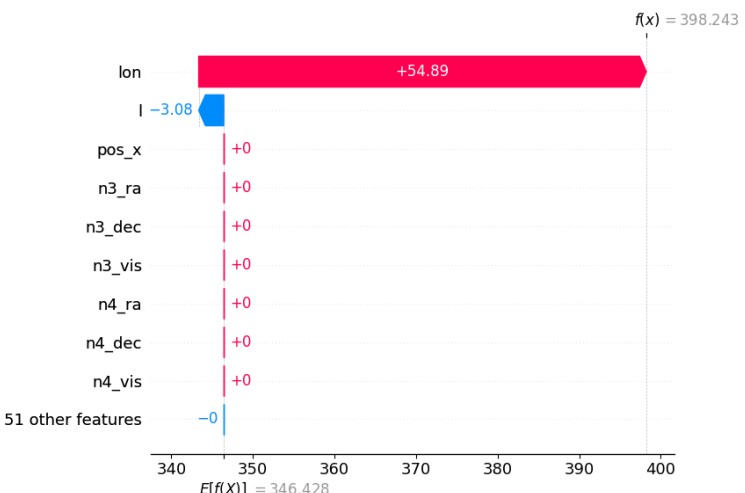

**Figure 7.** Kernel SHAP feature importance for the peak datapoint. The spacecraft geographical longitude is the most critical feature in determining the sudden count rate increase from a baseline count rate of 346 to 398 (the peak). This baseline is constructed using 15 datapoints before and after the peak, which are used for sampling values and perturbing the datapoint to provide an explanation.

Another significant error type, more severe due to its contribution to false positive detections during the 2014 period (characterized by the solar maximum phase with numerous and significant solar flares), involves sudden steps in the background estimation. Figure 8 illustrates this behavior for detector n8 in energy range r1. In the SHAP analysis (refer to Figure 9), the most influential features are related to a flag feature: 1 indicates the detector is visible, while 0 signifies that its FoV is obscured by the Earth. Interestingly, what stands out is that detector n8 behavior seems influenced by the visibility of other detectors. This peculiar pattern cannot be attributed to data errors, as their correctness has been verified, but rather to a learned pattern within the neural network. Figure 8 presents the values of these features, demonstrating a clear correlation between the step behavior and changes in these flags.

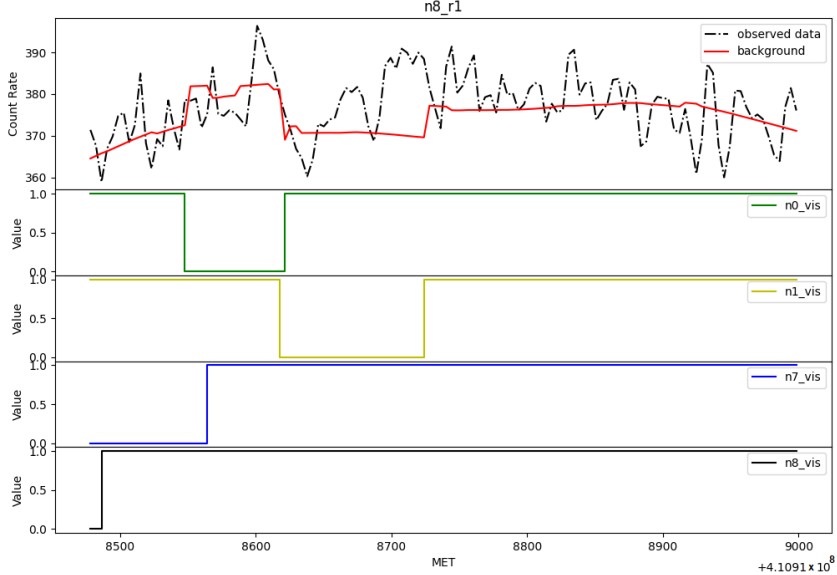

**Figure 8.** Observed and estimated count rate light curve for detector n8 in energy range *r*1. The unexpected steps in estimated count rates are attributed to variations in features such as n0_vis, n1_vis and n7_vis, with n8_vis remaining constant during this unusual estimation period.

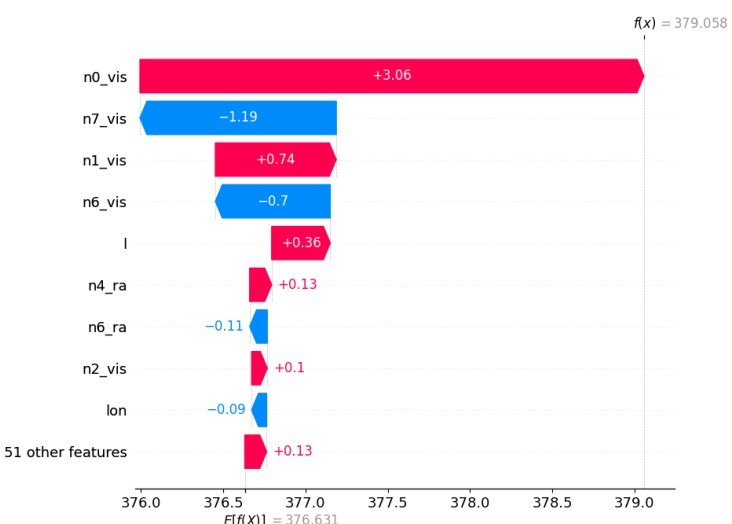

**Figure 9.** Kernel SHAP feature importance for a datapoint in a step of the estimated background in Figure 8. Notably, the most influential features are those related to detector visibility, specifically whether the FoV is obscured by the Earth or not. The baseline for this method is established by using 15 datapoints before and after the datapoint to be explained. These datapoints are used for value sampling and perturbation of the datapoint during the explanation process.

A potential simple solution in this scenario could involve omitting these features as inputs and evaluating whether model performance remains consistent and this behavior subsides. However, it is worth noting that these features might serve a purpose in correcting for solar activity effects. This contrasts with the 2019 period, during which FoV occultation was deemed irrelevant in background estimation (see Figure 4). Moreover, it is noted that the impact of occultation on detectors facing the Sun (from n0 to n5) causes a decrease in estimated count rates. In contrast, detectors from n6 to nb exhibit behavior similar to that depicted in Figure 8.

Another potential solution is integrating these features in the last layer only if it associated with the detector (e.g., n8_$r$1 count rate and n8_vis).

### 4. Summary and Conclusions

The detection of GRB transients with spaceborne photon detectors requires a accurate estimate of background count rates. Our study takes a data-driven approach, using a neural network that adapts to various spaceborne X/gamma-ray photon detectors to produce a reliable estimate of background count rates.

Quantile regression played a crucial role in our methodology, offering a nuanced estimation by providing a confidence range for predictions. Notably, the 90th percentile from the quantile regression emerged as a robust estimate, enhancing the reliability of our count rate predictions.

The neural network, designed for multi-output regression, showcased commendable performance across various training periods. We observed that the model achieved stability and optimal performance within a 4-month training period, with no substantial improvement observed beyond this duration.

Crucially, our experiments extended beyond the conventional in-time testing. We evaluated the NN performance on out-of-time test sets, simulating scenarios where the model encountered data outside its training period. The results demonstrated a level of generalization that, while not perfect, remained satisfactory, underscoring the model adaptability and reliability. Nonetheless, this emphasizes the importance of retraining the model for each period to achieve better estimates.

To enhance our understanding of the NN behavior, we employed Explainable Artificial Intelligence (XAI) techniques, including Morris sensitivity analysis and Kernel SHAP.

The global explanation through Morris sensitivity analysis unveiled the significance of various features, including the McIlwain L-parameter, geographical coordinates and detector flags indicating the Earth's Field of View (FoV) occultation. It was observed that the importance of these features could vary between different periods, emphasizing the need for a tailored neural network for each period, as demonstrated in 2014 and 2019. On the other hand, local explainability using Kernel SHAP assisted in understanding and, to some extent, debugging the neural network, leading to data fixes or suggestions, such as incorporating temporal features and investigating the omission of specific variables, to enhance the overall performance of the machine learning model.

In conclusion, our developed neural network-based background estimator holds promise for the robust detection of GRBs and other transient phenomena in high-energy space telescope observations. The data-driven background estimator, featuring robust techniques like quantile regression and XAI, contributes to the creation of adaptable models for offline analysis of astronomical data. As we look to the future, XAI could pave the way for enhanced model performance and a deeper understanding of the underlying processes.

**Author Contributions:** Conceptualization, R.C., F.F.; methodology, R.C.; software, R.C.; validation, R.C.; formal analysis, R.C.; investigation, R.C.; resources, F.F.; data curation, R.C.; writing—original draft preparation, R.C., G.D., G.D.C. and A.V.; writing—review and editing, R.C., G.D., G.D.C., F.F. and A.V.; supervision, A.V.; project administration, R.C. All authors have read and agreed to the published version of the manuscript.

**Funding:** This research received no external funding.

**Data Availability Statement:** The data employed in the present analysis were sourced from the Fermi GBM FTP site, accessible at https://heasarc.gsfc.nasa.gov/FTP/fermi/ (accessed on 20 January 2023).

**Acknowledgments:** We are grateful to Maria Dainotti for her insightful feedback provided during the examination of Riccardo Crupi's PhD thesis, which served as inspiration for the development of this paper.

**Conflicts of Interest:** The authors declare no conflicts of interest.

## Abbreviations

The following abbreviations are used in this manuscript:

| | |
|---|---|
| GRB | Gamma-Ray Burst |
| NN | Neural network |
| Fermi GBM | Fermi Gamma Burst Monitor |

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
