# Peer review of "Enhancing Gamma-Ray Burst Detection: Evaluation of Neural Network Background Estimator and Explainable AI Insights"

_galaxies, doi:10.3390/galaxies12020012_

Round 1

Reviewer 1 Report

Comments and Suggestions for Authors

This manuscript addresses a very challenging issue of large current interest; namely the identification and characterization of faint gamma ray bursts (GRBs). These are thought to result from mergers of stellar mass black holes and/or neutron stars or the core collapse of massive stars. Although luminous, the events are typically remote, so faint. Also gamma ray detections must be done above the atmosphere and are sensitive to solar activity. This manuscript offers an AI approach to reliable background removal under these circumstances. The approach seems valid and is clearly motivated and outlined.  There are a few "typo level" wording issues that should be addressed before the manusript is published.

Reviewer 2 Report

Comments and Suggestions for Authors

In their paper, the authors introduce a novel approach based on Neural Network techniques for estimating the background rate of spaceborne X/gamma-ray photon detectors. The paper is interesting and well written. However, I have identified some minor points, outlined below, which I recommend the authors address before finalizing the publication.

Section 2.1: Table 3 is presented prior to Tables 1 and 2. Please arrange the tables in accordance with their order of appearance.

Lines 73-74: The variables corresponding to "col_det_pos" are not clearly defined. Provide clarification on this matter.

Table 1: In the text, elaborate on the nature of the feature labeled "l."

Line 83: The statement "instances in South Atlantic Anomaly were deleted" lacks clarity. Please provide a more detailed explanation of what the South Atlantic Anomaly entails.

Line 114: The term "MeAE" should be defined and briefly described in the text.

Lines 126-128: The basis for deeming performance satisfactory is unclear. The authors should either elucidate the criteria or consider a more quantitative approach, such as expressing the effect as a percentage.

Line 143: The term "XAI technique" requires a more detailed explanation. Define XAI techniques in this context.

Subsection "global explanation": Address the discrepancy in results between Figure 4 and Figure 5. Offer an explanation for this divergence.

Subsection "local explanation": Rearrange the order of Figures 6 and 7, as Figure 7 is discussed first in the text.

Line 172: Define the term "longitudinal feature."

Lines 181-184: Similar to the issue with Figures 6 and 7, reconsider the order of Figures 9 and 8 in the text.
